# Cost-efficient Knowledge-based Question Answering with Large Language Models

**Junnan Dong**[1], **Qinggang Zhang**[1], **Chuang Zhou**[1], **Hao Chen**[1†], **Daochen Zha**[2], **Xiao Huang**[1]

[1] The Hong Kong Polytechnic University
[2] Rice University
{hanson.dong, qinggangg.zhang, chuang-qqzj.zhou}@connect.polyu.hk,
sundaychenhao@gmail.com,daochen.zha@rice.edu, xiaohuang@polyu.edu.hk

## Abstract

Knowledge-based question answering (KBQA) is widely used in many scenarios that necessitate domain knowledge. Large language models (LLMs) bring opportunities to KBQA, while their costs are significantly higher and absence of domain-specific knowledge during pre-training. We are motivated to combine LLMs and prior small models on knowledge graphs (KGMs) for both inferential accuracy and cost saving. However, it remains challenging since accuracy and cost are not readily combined in the optimization as two distinct metrics. It is also laborious for model selection since different models excel in diverse knowledge. To this end, we propose Coke, a novel cost-efficient strategy for KBQA with LLMs, modeled as a tailored multi-armed bandit problem to minimize calls to LLMs within limited budgets. We first formulate the *accuracy expectation* with a cluster-level Thompson Sampling for either KGMs or LLMs. A context-aware policy is optimized to further distinguish the expert model subject to the question semantics. The overall decision is bounded by the *cost regret* according to historical expenditure on failures. Extensive experiments showcase the superior performance of Coke, which moves the Pareto frontier with up to 20.89% saving of GPT-4 fees while achieving a 2.74% higher accuracy on the benchmark datasets.

## 1 Introduction

Knowledge-based question answering (KBQA) has gained significant attention across various specialized domains, e.g., education and medicine [24, 13, 21, 20]. Given a question, the model is required to make inferences based on necessary reasoning background [14]. Inspired by the effectiveness of knowledge graphs (KGs), e.g., ConceptNet [33, 8], where real-world entities are represented in the structural form as (*head entity*, *relation*, *tail entity*) [7], KG-based models (KGMs) have been proposed to leverage KGs for reasoning. Based on the hypothesis that answers could be located multiple hops away from the question concepts [24] in KGs, former studies are mainly dedicated to tracing the trajectory from questions to answers to model the structural information [16, 25, 5, 4], or utilizing graph neural networks (GNNs) to learn the question-specific subgraph from KGs [14, 37, 39].

With the emergence of large language models (LLMs), e.g., ChatGPT and GPT-4 [29], They have shown remarkable performance benefited from the injected knowledge during pre-training [38, 12]. However, it is challenging to adopt LLMs in practice. First, either calling the API or deploying the open-source LLMs with cloud service is prohibitive [2, 9]. GPT-4 is estimated to cost at least thousands of dollars for pilot-scale customer service [6] while Llama3 70B requires unaffordable

---

† Corresponding Author.

38th Conference on Neural Information Processing Systems (NeurIPS 2024).

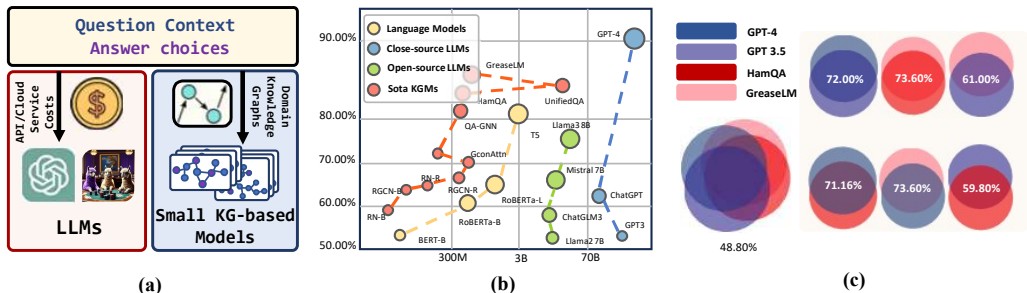

Figure 1: A sketched overview of LLMs and small KGMs in (a) We visualize the Acc./Param size of both pipelines of models in (b) The overlaps among different model predictions are shown in (c).

computation resources for small business, e.g., 40G graphics memory, let alone the high throughput scenarios like online e-commerce [30]. Second, LLMs may struggle to identify the correct answer for certain questions due to the lack of particular knowledge that is not covered in their pre-training corpus [1, 32, 15]. They are considered unreliable to assist pedagogical or medical purposes since they could generate hallucination [3, 18] and misleading responses [17, 19].

We illustrate the sketched overview of LLMs and KGMs methods in Figure 1 (a) where LLMs are used in a zero-shot setting with direct prompts, e.g., [Question:Choices], and KGMs require the external domain KG as reasoning background. In Figure 1 (b), we visualize the accuracy/parameter size of four types of models. In general, KGMs are far more lightweight considering model size but tend to underperform LLMs overall. Conversely, LLMs are more computationally expensive and may struggle with specific questions that demand knowledge not covered in the pre-training corpus. We are thereby motivated to combine the strengths of LLMs and KGMs through question-specific model selection to improve the Pareto frontier, achieving higher inferential accuracy and lower costs.

However, it is nontrivial for two major challenges. First, inferential accuracy and cost saving are two distinct metrics. It is hard to combine them in the optimization simultaneously. Since more parameters will lead to higher costs, it indicates the importance of a careful selection that balances both exploration and exploitation. Second, selecting the most suitable model for particular questions is laborious. In Figure 1 (c), we showcase the overlaps among several representative models on the benchmark OpenBookQA [28] dataset. While different models focus on various considerations, they may consequently excel in diverse knowledge and question types. For example, HamQA [14] focuses on hypernymy in the questions, i.e., cats and mammals. This makes it expensive to incorporate expertise according to the specialty of models to answer diverse real-world domain-specific questions.

To this end, we present a novel cost-efficient strategy to leverage LLMs for KBQA, i.e., Coke. It is modeled as a tailored multi-armed bandit (MAB) problem, which is trained to assign the most promising model for each question within a considerably limited budget. Specifically, we assemble two sets of base models, i.e., LLMs and KGMs to balance both inferential accuracy and cost saving. $(i)$ we first model the preliminary selection with a cluster-level Thompson Sampling. It suggests the *accuracy expectation* of choosing either LLMs or KGMs based on historical success and failure at Beta distribution. $(ii)$ A context-aware policy is learned to further distinguish the most suitable model. This could effectively assign the corresponding expert for the given question semantics. $(iii)$ The overall decision making is bounded by the *cost regret*. It indicatively constrains the selection based on the cumulative expenses incurred from failures of the current model.

**Contributions**.

▶ We formally define the task of optimizing both inferential accuracy and cost for KBQA with LLMs.

▶ A novel cost-efficient strategy, Coke, is presented to automatically assign the most promising model for particular questions. It could effectively make reliable decisions considering both *inferential accuracy* and *cost saving*, making a balance of *exploration* and *exploitation* during selections.

▶ Extensive experiments over three domain-specific benchmark datasets demonstrate the superiority of our proposed framework, moving the Pareto frontier to achieve higher accuracy and lower costs.

## 2 Problem Formulation

### 2.1 Task Definition

We adopt lowercase alphabets for scalars (e.g., $m$), boldface lowercase letters for vectors(e.g., $\mathbf{q}$) and boldface uppercase ones for matrices (e.g., $\mathbf{Q}$). The decorated letters are used for sets, e.g., $\mathcal{A}$. We assemble two clusters of models, i.e., $\mathcal{C} = \{c_L, c_K\}$ for sets of LLMs and KGMs, respectively. All the candidate models in $\mathcal{C}$ are denoted as $\mathcal{M} = \{m_1, m_2..m_n\}$. The knowledge graph used in KGMs is expressed as $\mathcal{G}$ where real-world entities $e$ are represented in the form of triples as $(e_h, r, e_t)$.

> Given domain-specific questions $Q = \{q_1, q_2..q_u\}$, and several candidate models $\mathcal{M}$, we aim to optimize the framework to identify the most promising model within a limited budget $\mathcal{B}$ to invoke LLMs. The overall performance is evaluated by both inferential accuracy and cost saving. We aim to maximize the accuracy comparing prediction $\hat{y}_i$ and ground truth $y_i$, i.e., $max(f_{acc}(\hat{y}_i|y_i))$, and minimize the costs, i.e., $min(f_{cost}(p(m)|q_i))$ where $p(m)$ is the unit cost of model $m$.

### 2.2 Performance Evaluation

For fair comparisons, we divide the overall evaluation of `Coke` against state-of-the-art baselines into two parts considering both *inference accuracy* and *cost saving*.

**Inferential accuracy:** The performance of KBQA task itself is evaluated by the overall accuracy of the model prediction compared with ground truths, i.e., $\{0, 1\}$ for each question indicates wrong/correct. The accuracy is expected to be as high as possible to correctly answer more questions.

**Cost Saving:** The cost of using particular models is often case by case. It could involve many aspects, e.g., power consumption, GPU and graphics memory costs, cloud service charges and token-level API expenses, etc. In this paper, we instantiate this metric for `Coke` in two ways; one can define the cost in other ways under our framework. $(i)$ *API fees* (\$). This intuitively indicates the cost of money particularly for comparing with a series of LLMs from OpenAI, e.g., GPT3.5 and GPT4. $(ii)$ *Calls* (times). We generalize the evaluation to open-source LLMs like Llama and ChatGLM. It indicates the number of times that we invoke the LLMs. Since KGMs are considered far more cheaper for local and cloud implementation, the metric of calls is also expected to be as few as possible.

## 3 Approach: `Coke`

To achieve an effective model selection, we mainly aim to answer two research questions: $(i)$ ***How can we balance the exploration and exploitation to find the best model for both accuracy and cost saving?*** Real-world questions are difficult to manually identify to apply LLMs or KGMs. We usually value the prior knowledge to select the most accurate and inexpensive model at present greedily, i.e., *exploitation*. However, we also wish to obtain sufficient posterior experiences from under-explored models so far, i.e., *exploration* to find more models with a superior performance-to-price ratio. $(ii)$ ***How can we automatically distinguish the most promising expert models for different questions?*** The particular types of required knowledge vary among questions. This suggests a careful selection to leverage the specialized expertise from different models to make correct inferences. To this end, we design a novel cost-efficient framework, `Coke`, to automatically select question-specif models.

We formulate the automatic model selection as a multi-armed bandit (MAB) problem. Specifically, in each iteration $k \leq \mathcal{K}$, our policy is presented with the choice of selecting one model $m \in \mathcal{M}$ out of $\mathcal{N}$ candidate models, referred to as $\mathcal{N}$ *arms*. Let $r_m^k \in \{0, 1\}$ denote the reward obtained by selecting the model $m$ at the iteration $k$, where $m \in \mathcal{M}$. For each given question $q$, if the chosen model $m$ can correctly answer it, the policy will receive a positive reward of 1, and 0 vice versa. We aim to maximize the cumulative rewards in $\mathcal{K}$ iterations and formulate the objective as follows:

$$\max \sum_{k=1}^{\mathcal{K}} r_m^k. \tag{1}$$

In each iteration, the selection is inherently associated with an expectation function $\mathbb{E}(\cdot)$ to capture the implicit linear correlation between the given question $q$ and the potential reward $r$, indicating the likelihood of success by choosing one particular model $m$.

## 3.1 Accuracy-Cost Trade-off for KBQA with LLMs

To answer the two aforementioned questions, we decompose the expectation $\mathbb{E}(r_m^k|q)$ into three aspects. First, we calculate the *accuracy expectation* of one cluster, i.e., LLMs or KGMs, for better inferential performance(*). Second, we design a context-aware expert distinguishing to present the question embedding to the policy and find the best arm with implicit expertise (**), which further increases the probability of achieving higher accuracy for different question contexts. Finally, we constrain the decision-making with a *cost regret*. This is introduced to punish the model which wastes more on failures (***). The overall expectation is correspondingly formulated as follows.

$$\mathbb{E}(r_m^k|q_k) = \underbrace{\mathbb{E}_c(r_c|\theta_c)}_{(*)} + \underbrace{\mathbb{E}_a(r_a|q_k)}_{(**)} - \lambda \cdot \underbrace{\mathcal{R}_a(\mathcal{B}, p(a), q_i)}_{(***)}, \tag{2}$$

where $r_m^k \in \{0, 1\}$, $r_c$ and $r_a$ are the decomposed rewards for cluster sampling in terms of accuracy and arm selection considering knowledge expertise for particular questions. $\mathcal{B}$ indicates the limited budget allocated to invoke LLMs. Details are described hereunder.

## 3.2 Accuracy-Encouraged Cluster Expectation (*)

To encourage the policy to select more promising models, we first establish a higher-level expectation concerning the overall accuracy performance of KGMs and LLMs. Inspired by the traditional Thompson Sampling, which iteratively samples the best arm based on historical information, i.e., success and failure, we thereby design a tailored cluster-level Thompson Sampling to evaluate the expectation of choosing one particular cluster $c$. It presents a dynamic approach where the selections evolve over iterations based on success and failures, gradually converging towards optimal selections as more questions are encountered. This could also effectively embody the *exploration-exploitation* trade-off inherent in our model selection scenarios for the sake of accuracy.

Specifically, it is established based on the prior knowledge as *(success, failure)* of each cluster, instantiated by a Beta distribution of $Beta(\alpha, \beta)$. We define this pair of conjugate prior below.

**Definition: conjugate prior of $\alpha$ and $\beta$.** *Given a cluster $c \in \{c_L, c_K\}$, let $\alpha_c$ and $\beta_c$ denote the success and failure prior respectively for c. The pair of $(\alpha_c, \beta_c)$ forms a conjugate prior. It facilitates the efficient update of fundamental beliefs about the performance of each cluster c during selections.*

In general, we consider our cluster-level accuracy expectation $\mathbb{E}_c$ as a likelihood of success for each cluster by randomly sampling an indicator $\theta_c^k$ in iteration $k$ from the distribution of $Beta(\alpha_c^{k-1}, \beta_c^{k-1})$ based on the observation of success and failure in the former $(k-1)$ rounds.

$$\mathbb{E}_c(r_c^k|\theta_c) \propto \theta_c \sim (\alpha_c^{k-1}, \beta_c^{k-1}), \tag{3}$$

where $\theta_c$ is distributed approximately uniformly across the entire interval, resulting in a uniform *exploration* when a particular cluster has not been extensively sampled. Otherwise, if cluster $c$ has been sufficiently selected and the performance turns out to be satisfying with more success times, i.e., larger $\alpha_c$, the corresponding expectation associated with $\theta_c$ will be more likely to be higher to facilitate a reliable *exploitation*. When $k = 0$, we value the prior knowledge of cluster expectation before any observations have been made. In our paper, we utilize the average reported performance of each arm within the cluster as the prior for $\theta_c$. This ensures an empirically grounded initial belief about the success probability of the cluster, as well as for subsequent Bayesian inference.

$$prior(\theta_c^k) \sim \frac{\Gamma(\alpha_c^{k-1} + \beta_c^{k-1})}{\Gamma(\alpha_c^{k-1}) \cdot \Gamma(\beta_c^{k-1})} \cdot \theta_c^{\alpha_c - 1} \cdot (1 - \theta_c)^{\beta_c^{k-1} - 1}, \tag{4}$$

where $\Gamma(\cdot)$ is the Gamma function. Based on this, we consider the cluster with the largest $\theta_c^k$ in iteration $k$ as the best cluster $c^*$ for current question $q$, where the arm models within this particular cluster will have higher chances of answering this question. A reward $r_c^k \in \{1, 0\}$ will then be given if $c^*$ can/cannot make the correct prediction. Correspondingly, the posterior distributions for all the historically selected clusters will be updated when $0 < k \leq \mathcal{K}$. The history of cluster sampling and the posterior updating is formulated as follows, respectively.

$$\mathcal{H}_c^{k-1} = \{c_n^\tau, \alpha_c^{k-1}, \beta_c^{k-1}, \mathbf{r}_c^{k-1}, \tau = 1, 2...k-1, n = 1, 2...N\}$$

$$posterior(\theta_c^k) \sim \frac{\Gamma(\alpha_c^{k-1} + \beta_c^{k-1} + 1)}{\Gamma(\alpha_c^{k-1} + r_c^{k-1}) \cdot \Gamma(\beta_c^{k-1} + 1 - r_c^{k-1})} \cdot (\theta_c^k)^{\alpha_c^{k-1} + r_c^{k-1} - 1} \cdot (1 - \theta_c^k)^{\beta_c^{k-1} - r_c^{k-1}}. \tag{5}$$

While the exact sequence of cluster selections may vary between runs, the overall behavior will eventually converge to optimal cluster selections over time in our modeled MAB problem, especially as more questions are presented and more historical success/failure information is observed.

**Definition: Selection Regret.** *In iteration $k$, we denote the real-selected arm as $a_k$, the annotated best arm as $a_k^*$, which can answer the question with the lowest costs. We refer to the expectation differences between $a_k$ and $a_k^*$ as the selection regret for current iteration $k$.*

Specifically, we give the overall selection regret bounds for the cluster-level Thompson Sampling as follows. Given the selected arm $a_k$ and the historical information $H_k$ up to iteration $k$, $\mathbb{E}_c$ is bounded as follows: (The detailed proof of confidence bound is provided in the **Appendix** Section A)

$$SR(\mathcal{K}) \leq 2\gamma + 2\sum_{k=1}^{\mathcal{K}} \mathbb{E}[r(a_k, H_{k-1})]. \tag{6}$$

In general, the bounds are obtained from the following derivations. Initially, we establish the upper and lower confidence bounds as explicit functions of the arm $a_k$ and history $H_{k-1}$ respectively, denoted as $U(a_k, H_{k-1})$ and $L(a_k, H_{k-1})$. For some $\gamma > 0$ and the number of arms $\mathcal{N}$, the specific form of functions $U$ and $L$ is irrelevant as long as they satisfy the following properties:

$$\forall a, k, \quad \mathbb{E}\big[[U(a, H_{k-1}) - \mu(a)]^-\big] \leq \frac{\gamma}{\mathcal{K} \cdot \mathcal{N}},$$
$$\forall a, k, \quad \mathbb{E}\big[[\mu(a) - L(a, H_{k-1})]^-\big] \leq \frac{\gamma}{\mathcal{K} \cdot \mathcal{N}}. \tag{7}$$

### 3.3 Context-Aware Expert Distinguishing (**)

To answer the second question, we further deepen our MAB problem as a contextual variant. In this part, the expectation is highly related to the question semantics. We aim to automatically learn from the vector representation of questions, e.g., $\mathbf{q} \in \mathbb{R}^d$, $d$ for dimension, and effectively identify the corresponding expert model to answer it. The embedding is uniformly obtained by applying a lightweight pre-trained language model RoberTa [27]. To this end, we design the expectation function $\mathbb{E}_a(r_a^k | q^k)$ to effectively capture the linear correlation between $\mathbf{q}$ and $r_a$ in iteration $k$.

$$\mathbb{E}_a(r_a^k | q^k) = \mathbf{q}^k \times \boldsymbol{\mu}_a^{k-1} + \eta_a^{k-1}, \tag{8}$$

where $\boldsymbol{\mu}_a^{k-1} \in \mathbb{R}^{1 \times d}$ is a learned vectored parameter associated with each arm $a$ in $k-1$ steps. We introduce $\eta_a^{k-1}$ as a trainable noise at Gaussian distribution $\hat{\mathcal{N}}(0, (\Delta^{(n)})^2)$ to balance the *exploration* and *exploitation*. We maximize $\mathbf{q}^k \times \boldsymbol{\mu}_a^{k-1}$ to encourage the exploitation [10]. The tight correlations are established among the given question, the history information, i.e., $\mathcal{H}_a^{k-1}$, including all the questions $\mathbf{Q}_a^{k-1} \in \mathbb{R}^{(k-1) \times d}$ answered by arm $a$ and the rewards received $\mathbf{r}_a^{k-1} \in \mathbb{R}^{1 \times (k-1)}$ in k-1 iterations. We could observe the history $\mathcal{H}$ for each arm for abundant information for reference as:

$$\mathcal{H}_a^{k-1} = \{a_n^\tau, \mathbf{Q}_a^{k-1}, \mathbf{r}_a^{k-1}, \tau = 1, 2...k-1, n = 1, 2...N\} \tag{9}$$

Specifically, we update $\boldsymbol{\mu}_a^k$ based on the $\mathcal{H}_a^{k-1}$ with a typical ridge regression $f(\cdot)$.

$$f(\mathbf{Q}_a^{k-1}, \mathbf{r}_a^{k-1}) = \sum_{k=1}^{K} (\mathbf{r}_a^{k-1} - \mathbf{Q}_a^{k-1}\boldsymbol{\mu}_a^k)^2 + \sigma \parallel \boldsymbol{\mu}_a^k \parallel_2^2$$
$$\rightarrow f(\boldsymbol{\mu}_a^k) = (\mathbf{r}_a^{k-1} - \mathbf{Q}_a^{k-1}(\boldsymbol{\mu}_a^k)^\top)(\mathbf{r}_a^{k-1} - \mathbf{Q}_a^{k-1}\boldsymbol{\mu}_a^k) + \sigma^b (\boldsymbol{\mu}_a^k)^\top \boldsymbol{\mu}_a^k, \tag{10}$$

where we introduce the L2 normalization to ensure the reversibility of $\mathbf{Q}_a^{k-1}$ in addition to the original ordinary least square loss. $\sigma$ solves the over-fitting problem by adopting a suitable $\lambda$. To find the optimal value of $\boldsymbol{\mu}_a^k$ that minimizes the cost function, we differentiate the formula with respect to $\boldsymbol{\mu}_a^k$, set the derivative equal to zero, and solve it. This yields the following update equation:

$$\boldsymbol{\mu}_a^k = \big((\mathbf{Q}_a^{k-1})^\top \mathbf{Q}_a^{k-1} + \sigma_a \mathbf{I}\big)^{-1} (\mathbf{Q}_a^{k-1})^\top \mathbf{r}_a^{k-1}, \tag{11}$$

where $\mathbf{I} \in \mathbb{R}^{d \times d}$ is an identity matrix. To facilitate exploration on less explored arms, we adopt an upper confidence bound for exploration-exploitation trade-off by introducing $\eta_a^{k-1}$. For any $\delta > 0$ with the probability at least $(1 - \delta)$, the expectation $\mathbb{E}_a(r_a^k | \mathbf{q}^k)$ is bounded by a confidence interval:

$$\mathbf{q}^k \boldsymbol{\mu} a^{k-1} - \gamma \times f\gamma(\mathbf{Q}_a^{k-1}) \leq \mathbb{E}_a \leq \mathbf{q}^k \boldsymbol{\mu} a^{k-1} + \gamma \times f\gamma(\mathbf{Q}_a^{k-1}), \tag{12}$$

where $\gamma$ is a constant value, i.e., $\gamma = 1 + \sqrt{ln(2/\delta)/2}$. Also, we can derive the correlation term $f\gamma(\mathbf{Q}a^{k-1})$ as $f\gamma(\mathbf{Q}_a^{k-1}) \triangleq \sqrt{\mathbf{q}^k \left((\mathbf{Q}_a^{k-1})^\top \mathbf{Q}_a^{k-1} + \sigma\mathbf{I}\right)^{-1} (\mathbf{q}^k)^\top}$. Hence, through learning from the current question $\mathbf{q}^k \in \mathbb{R}^d$ and all historically assigned questions $\mathbf{Q}_a^{k-1} \in \mathbb{R}^{(k-1)\times d}$ for arm $a$, we can simply derive the equation to appropriately update the noise term $\eta_a^{k-1}$ for next iteration.

$$\eta_a^k = \gamma \times \sqrt{\mathbf{q}^k \left((\mathbf{Q}_a^{k-1})^\top \mathbf{Q}_a^{k-1} + \sigma_a\mathbf{I}\right)^{-1} (\mathbf{q}^k)^\top}. \tag{13}$$

When an arm $a$ with larger $\mathbf{q}^k \boldsymbol{\mu}_a^{k-1}$ is selected, this reflects an *exploitation* process. Similarly, when the model chooses an arm with larger $\eta_a^{k-1}$ learned in previous iterations, this variance shows an *exploration* process since the model performs few selections of the current arm. Thus, jointly maximizing $(\mathbf{q}^k \times \boldsymbol{\mu}_a^{k-1} + \eta_a^{k-1})$ helps us for more promising expert models subject to question $q^k$.

In conclusion, guided by the objective of maximizing cumulative rewards $r_m^k$, we concentrate on the sub-target of finding contextually best arm as the expert to correctly answer the question by prioritizing the arm with higher expectation $\mathbb{E}_a(r_a^k|q^k)$ to obtain $r_a^k$.

### 3.4 Cost Regret Constraint (***)

Since the budget is limited, we aim to make the best use of the chances for both accuracy improvement and cost savings. In this part, we introduce a penalty term *cost regret*, denoted as $\mathcal{R}_a$, to measure the proportion of costs incurred by incorrect predictions given by the arm $a$ within a budget constraint.

$$\mathcal{R}_a = \frac{\sum_{q\in Q_a^\beta} p(a_k)\|q_a^\beta\|}{\sum_{q\in\{Q_a^\alpha \cup Q_a^\beta\}} p(a_k)\|q_a\|}$$
$$s.t. \sum_{q\in\{Q_a^{k-1}\}} p(a_k)\|q_a\| + p(a_k)\|Q_a^k\| \leq \mathcal{B}; \forall a \in \mathcal{A}. \tag{14}$$

where $q_a^\beta$ and $Q_a^\beta$ indicate the failure, i.e., the question and historically assigned questions for arm $a$ that are wrongly answered. $p(a)$ is the unit cost for invoking the LLMs. For black-box LLMs, we calculate $p(a)\|q_a$ as the token-level fees; while for open-sourced LLMs, it will be replaced by the times that we call LLMs. The numerator of $\mathcal{R}_a$ represents the total cost incurred by incorrect answers given by arm $a$, while the denominator calculates the total cost of all questions answered by arm $a$. The constraint could effectively ensure that the total cost of selecting arm $a$ for answering questions in the previous $k-1$ iterations and the current iteration $k$ does not exceed a predefined budget $\mathcal{B}$. To control the impacts of over-constraint from *cost regret* which may penalize the model for the costs on necessary trials, we introduce $\lambda$ as a hyperparameter to control the trade-off between maximizing rewards and minimizing cost regret for a reasonable *exploration* and *exploitation*.

## 4 Experiments

We conduct experiments on three domain-specific datasets: $(i)$ Commonsense knowledge domain: CommonsenseQA [35]; $(ii)$ Scientific Openbook domain: OpenbookQA [28]; $(iii)$ Medical Domain: MedQA-USMLE [23]. To compare the performance of `Coke`, we include the baselines from three aspects, i.e., fine-tuned Language Models, KGMs and both API series and local series of LLMs. Additionally, our framework is efficiently runnable on one CPU. To accelerate the matrix computation, we adopt Torch to boost the selection on an NVIDIA GeForce RTX 4090 GPU.

### 4.1 Datasets

**CommonsenseQA** [35] (abbreviated as *CSQA*) is a prominent dataset in the commonsense knowledge domain that demands extensive real-world commonsense knowledge. It encompasses 12,102 questions, and ConceptNet [33], one of the largest commonsense knowledge graphs (KG), is frequently employed by existing KGMs for reasoning. Due to the official test set being reserved for leaderboard evaluations, we assess model performance using the in-house (IH) data split as implemented in [25]. **OpenBookQA** [28] (referred to as *OBQA*) is a scientific domain dataset that comprises 5,957 multiple-choice questions from open book exams, each with four options. Answering *OBQA* questions necessitates a comprehensive understanding of fundamental science facts and their applications, which involves understanding scientific principles and applying them to novel situations.

Table 1: Performance comparison among state-of-the-art baselines and `Coke` on three benchmark datasets in terms of both inferential accuracy and cost saving ($ API fees).

| Model | CommonsenseQA | | OpenBookQA | | MedQA | |
|---|---|---|---|---|---|---|
| | IHdev-Acc. | IHtest-Acc. | Dev-Acc. | Test-Acc. | Dev-Acc. | Test-Acc. |
| **Fine-dtuned Language Models (LMs)** | | | | | | |
| Bert-base [11] | 0.573 | 0.535 | 0.588 | 0.566 | 0.359 | 0.344 |
| Bert-large [11] | 0.611 | 0.554 | 0.626 | 0.602 | 0.373 | 0.367 |
| RoBerta-large [27] | 0.731 | 0.687 | 0.668 | 0.648 | 0.369 | 0.361 |
| **Knowledge Graph Based Small Models (KGMs)** | | | | | | |
| MHGRN [16] | 0.745 | 0.713 | 0.786 | 0.806 | - | - |
| QA-GNN [37] | 0.765 | 0.733 | 0.836 | 0.828 | 0.394 | 0.381 |
| HamQA [14] | 0.769 | 0.739 | 0.858 | 0.846 | 0.396 | 0.385 |
| JointLK [34] | 0.777 | 0.744 | 0.864 | 0.856 | 0.411 | 0.403 |
| GreaseLM [39] | **0.785** | 0.742 | 0.857 | 0.848 | 0.400 | 0.385 |
| GrapeQA [36] | 0.782 | 0.749 | 0.849 | 0.824 | 0.401 | 0.395 |
| **Large Language Models (LLMs) - Local Series** | | | | | | |
| ChatGLM | 0.473 | 0.469 | 0.352 | 0.360 | 0.346 | 0.366 |
| Baichuan-7B | 0.491 | 0.476 | 0.411 | 0.395 | 0.334 | 0.319 |
| Llama2 (7b) | 0.561 | 0.547 | 0.526 | 0.466 | 0.302 | 0.299 |
| Llama3 (8b) | 0.754 | 0.720 | 0.762 | 0.756 | 0.622 | 0.691 |
| `Coke` (Ours) | - | - | - | - | **0.627** | **0.692** |
| Acc. Imp.% vs. Llama3 (8b) | - | - | - | - | + 0.81% | + 0.16% |
| Cost Sav. (calls) % vs. Llama3 (8b) | - | - | - | - | - 1.17% | - 0.66% |
| **Large Language Models (LLMs) - API Series** | | | | | | |
| GPT3 | 0.539 | 0.520 | 0.420 | 0.482 | 0.312 | 0.289 |
| GPT3.5 | 0.735 | 0.710 | 0.598 | 0.600 | 0.484 | 0.487 |
| GPT-4 | 0.782 | 0.802 | 0.898 | 0.902 | 0.739 | 0.770 |
| `Coke` (Ours) | **0.802** | **0.824** | **0.902** | **0.908** | **0.746** | **0.778** |
| Acc. Imp.% vs. GPT-4 | + 2.56% | + 2.74% | + 1.12% | + 0.67% | + 0.95% | + 1.03% |
| Cost Sav. ($) % vs. GPT-4 | - 15.14% | - 20.89% | - 5.33% | - 11.02% | - 2.11% | - 4.32% |

**MedQA-USMLE** [23], i.e., *MedQA*, serves as a domain-specific question-answering benchmark focused on medical knowledge. This dataset is derived from the United States Medical Licensing Examination, which is a comprehensive and challenging assessment used to evaluate the competence of prospective medical professionals. MedQA includes a variety of question types that test clinical knowledge, diagnostic reasoning, and medical science applications. The dataset benchmarks the performance in understanding and applying medical knowledge in both healthcare and medicine.

## 4.2 Baselines

**Fine-tuned Language Models** abbreviated as LMs. We evaluate our method against standard fine-tuned language models, specifically using Bert-base, Bert-large [11], and RoBerta-large [27].
**KGMs** We include off-the-shelf small KGMs that integrate KGs for KBQA, including MHGRN [16], QA-GNN [37], HamQA [14], JointLK [34], GreaseLM [39], and GrapeQA [36].
**LLMs** Our comparison includes two categories of LLMs: the API series (GPT-3, GPT-3.5, and GPT-4) and local series (ChatGLM, Baichuan-7B, Llama2 (7b), and Llama3 (8b).

## 4.3 Main Results

The overall performance is compared and shown in Table 1. To compare with API series of LLMs, we assemble HamQA, GPT3.5 and GPT-4 as the arms and our proposed `Coke` has outperformed all four categories of baselines in terms of inferential accuracy. Specifically, we could achieve 2.03%, 0.67% and 1.03% improvements over GPT-4 on three datasets. For the local series of LLMs, we

adopt HamQA, Llama2 (7b) and Llama3 (8b). Since Llama2 (7b) and Llama3 (8b) underperform the traditional KGMs with lagging performance on both CSQA and OBQA, we merely conduct the experiments on MedQA with the arm models, i.e., HamQA, Llama2 (7B) and Llama3 (8B). We conclude our observations hereunder. The performance gap among different arms plays a vital role in balancing the accuracy and costs. For example, on CSQA and OBQA, the accuracy of state-of-the-art KGMs are very close to GPT-4 and much better that both GPT 3.5 and local series of LLMs. This facilitates a big improvements on cost saving by invoking more KGMs, where we achieve higher accuracy with much lower costs, i.e., 20.89% and 11.02%. However, on MedQA, where the questions are over-complicated and open-ended for KGMs to infer, there exists a huge performance gap between the best KGM and all the LLMs, especially when compared with GPT-4. Our policy has to rely on sufficient calls of LLMs to ensure accuracy, which indeed increases the costs.

## 4.4 Hyperparameter Analysis

In this subsection, we conduct a detailed analysis of the important hyperparameters, i.e., $\lambda$ and $\mathcal{B}$.

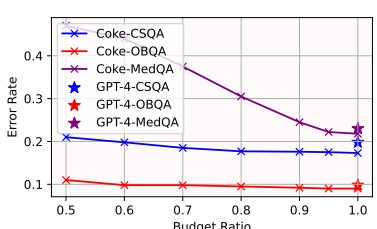

Figure 2: A visualization of Pareto frontier of both inferential accuracy and cost saving as budget $\mathcal{B}$ increases on three datasets.

### 4.4.1 Budget Study

The Pareto frontier for both inferential accuracy and cost saving is shown in Figure 2. For clear illustration, we replace the accuracy with the error rate which reversely shows the direction of performance changes. To observe when the accuracy and cost reach the balance, we decrease the budget from 1 to 0.5 until `Coke` has a higher error rate than GPT-4 $\mathcal{B} \in \{0.5, 0.6, 0.7..., 1\}$. In this figure, nodes in the lower left positions imply better performance considering both higher accuracy and lower costs. Specifically, our proposed `Coke` could achieve comparable performance to GPT-4 within around 60% budget on both CSQA and OBQA datasets. On MedQA, the slope rapidly drops before the budget ratio reaches 95% and finally outperforms GPT-4 after $\mathcal{B} \geq 0.95$. Since budget $\mathcal{B}$ strictly constrains the opportunities to call LLMs, intuitively, more budgets for LLMs will lead to higher accuracy in most scenarios. However, this does not practically stand for KBQA since the domain knowledge in LLMs is limited. Moreover, purely relying on LLMs will also lose the opportunities to leverage KGMs to further boost the accuracy. On the other hand, more budget will inevitably increase the token-level costs. Consequently, our proposed `Coke` has outperformed GPT-4 on all the datasets with higher accuracy as $\mathcal{B}$ increases. It effectively moves the Pareto frontier for KBQA with LLMs.

## 4.5 Observations on search of $\lambda$

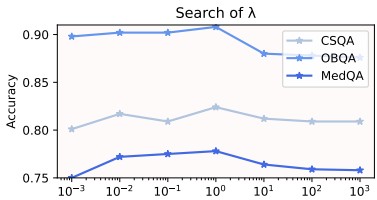

Figure 3: Performance changes based on the search of $\lambda$.

The hyperparameter $\lambda$ is essential for controlling the constraint from cost regret within our framework. We face a dilemma to cautiously decide the value of $\lambda$. On one hand, we aim to adopt a large value to strictly penalize models that allocate more resources to failed attempts. On the other hand, an over-penalty will damage the exploration during selection, which means the penalized models will no longer be invoked. Thus, we carefully adjust $\lambda$ within $\{0.001, 0.1, 1, 10, 100\}$ to modulate the extent to which we minimize cost regret versus maximizing accuracy. The visualization of performance, i.e., inferential accuracy and cost saving, is shown in Figure 3. Consequently, we conclude our observations as follows. A higher $\lambda$ value signifies a stronger emphasis on cost-efficient decision-making, potentially leading to more conservative model selections. Conversely, a lower $\lambda$ value may prioritize accuracy over cost savings, resulting in a higher tolerance for resource expenditure on exploration. We adopt the wave peak value of 1 for the final reported performance.

## 4.6 Ablation Studies

In this part, we investigate the importance of each decomposed expectation in our overall objective of optimization, i.e., $\mathbb{E}_c$ and $\mathbb{E}_a$, as well as the constraint from *cost regret* $\mathcal{R}_a$. The performance of

Table 2: Verification of the importance of $\mathbb{E}_c$, $\mathbb{E}_a$ and $\mathcal{R}_a$ on three datasets.

| Ablations | CommonsenseQA | | OpenBookQA | | MedQA | |
|---|---|---|---|---|---|---|
| | Accuracy | Cost Saving ($) | Accuracy | Cost Saving ($) | Accuracy | Cost Saving ($) |
| w/o $\mathbb{E}_c$ | 0.750 | - 47.59% | 0.855 | - 32.10% | 0.607 | - 30.51% |
| w/o $\mathbb{E}_a$ | 0.801 | - 15.42% | 0.880 | - 21.16% | 0.760 | - 1.07% |
| w/o $\mathcal{R}_a$ | 0.800 | - 6.26% | 0.898 | - 3.31% | 0.684 | - 15.98% |
| Coke | **0.824** | **- 20.89%** | **0.908** | **- 11.02%** | **0.778** | **- 4.32%** |

inferential accuracy and cost saving by removing one component are shown in Table 2. Removing $\mathbb{E}_c$ leads to significant reductions in cost savings across all datasets. Specifically, cost savings decrease by 47.59% for CSQA, 32.10% for OBQA, and 30.51% for MedQA. While The exclusion of $\mathbb{E}_a$ results in noticeable reductions in accuracy, especially for MedQA, where accuracy falls to 0.760. Finally, removing $\mathcal{R}_a$ brings decreases in both accuracy and cost savings across all datasets. The observations suggest the unique contribution of each component to the model's overall performance.

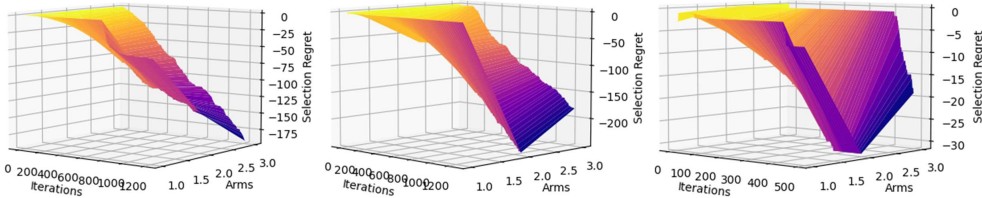

Figure 4: A 3D toy visualization of the selection regret on three datasets as iteration $k$ goes.

## 4.7 Selection Regret Analysis

Additionally to the proof of the expectation bounds of both $\mathbb{E}_c$ and $\mathbb{E}_a$, we comprehensively evaluate our model selection by visualizing the selection regret in a 3D figure across three datasets in Figure 4. For a clear demonstration of the performance changes, we instantiate the expectation gap $\mathbb{E}[\mu(a_k^*) - \mu(a_k)]$ between best arm $a_k^*$ and the selected arm $a_k$ with a toy example as {-1,0} which indicate the regret of choosing $a_k$. It sheds light on an intuitive understanding of how regret evolves and converges quickly over iterations, offering valuable insights into the model performance and the correctness of our selection strategy, highlighting the strengths of our proposed method.

## 4.8 Case Studies

To provide insights into the effectiveness of Coke on balancing inferential accuracy and cost saving, we visualize the distribution of model selection on CSQA, OBQA and MedQA in Figure 5, respectively. We could clearly observe the *exploration* process when the color of cubes in the heatmap changes from deep to shallow, e.g., {250, 500} intervals on CSQA and MedQA for GPT-4. This makes trials and spends necessary costs on the under-explored model. While the color changes from shallow to deep, e.g., {250, 500} intervals on CSQA for HamQA, {500, 750} intervals on CSQA and MedQA, {100, 200} on OBQA for GPT-4 and {750, 1000} intervals for ChatGPT, indicating the *exploitation* process that leverages the best model. The case study sheds light on our superior ability to balance the selection for more accurate and cost-saving candidate models.

## 5 Related Work

Contemporary research has been focused on generating answers through deductive processes applied to knowledge graphs, as outlined in surveys and studies on the subject [24, 22]. Existing methods predominantly leverage techniques rooted in semantic parsing and information retrieval [31, 26]. For instance, KagNet [25] introduces a schema graph that adeptly captures relational paths among pivotal entities. Nonetheless, these methodologies often delineate the processes of question interpretation and knowledge graph (KG) inference as distinct stages, thereby exposing the models to potential pitfalls associated with the nuanced or implicit facets of query phrasing, including negations and

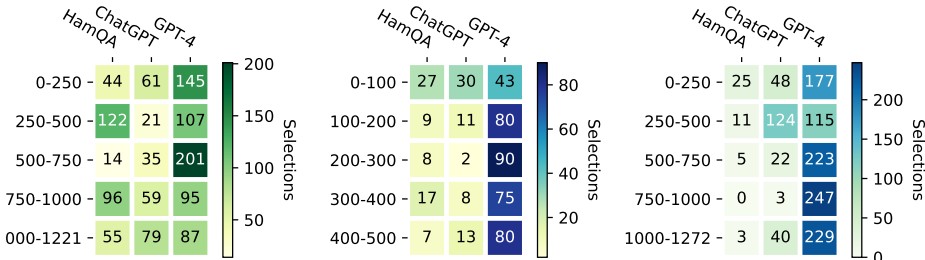

Figure 5: A case study of the model selection on three domain-specific datasets as $k$ goes. The color changes from deep to shallow indicates an *exploration* process, while an *exploitation* reversely.

contextual constraints. Recognizing this limitation, recent innovations have pivoted towards a more integrated approach to reasoning. MHGRN [16] exemplifies this trend by dynamically refining the embeddings of the question context in tandem with the reasoning process, utilizing graph neural networks. This fusion of path encoders with GNNs not only enhances the model's interpretability but also its ability to scale. Building on this, the QA-GNN framework [37] goes further by crafting a work graph wherein the context itself is instantiated as an entity and interlinked with relevant entities through cooccurrence, thereby streamlining the reasoning process. GreaseLM [39] establishes a tighter connection between language models and KGs with a joint training mechanism. While HamQA [14] focuses on combining question understanding and knowledge reasoning, it excels in answering hierarchical questions containing hyponymys.

## 6 Limitation

Our study faces a major limitation of the distinct performance gaps among existing models considering both inferential accuracy and cost savings, e.g., KGMs and LLMs. This disparity from the underperformance of certain models hinders the overall efficiency of model selection. For example, on MedQA dataset, the gap between the best KGM and GPT-4 is a remarkable 36.50%. This forces our policy to rely on GPT-4 to maintain accuracy, bringing higher costs. However, as a fast-pluggable policy, `Coke` can easily address this concern when more robust models appear to boost the selection.

## 7 Conclusion

In this paper, we present `Coke`, a novel cost-efficient strategy for LLMs in KBQA while balancing inferential accuracy and cost saving. Our work first formally defines the problem of trading off accuracy and cost for KBQA with LLMs and provides a practical solution for utilizing LLMs in resource-constrained and domain knowledge-required scenarios.`Coke` could effectively integrate two sets of off-the-shelf models, i.e., LLMs and KGMs, and efficiently assign the most promising model for each question within a limited budget by employing a tailored cluster-based Thompson Sampling and a contextual multi-armed bandit. The former models the preliminary selection between LLMs and KGMs based on historical performance, while the latter identifies the best model within a cluster according to question semantics. The overall decision-making is bounded by the cost regret, constraining the selection based on cumulative expenses incurred from model failures. Extensive experiments on three domain-specific benchmark datasets demonstrate the superiority of `Coke` in terms of both inferential accuracy and cost-effectiveness. Our proposed framework could also offer a significantly promising direction for efficient integration of LLMs in various knowledge-based tasks.

## Acknowledgement

The work described in this paper was fully supported by a grant from the Research Grants Council of the Hong Kong Special Administrative Region, China (Project No. PolyU 25208322).

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

## A  Proof of Selection Regret in Cluster-level Thompson Sampling

The expectation bounds of the specific iteration $t$ is:

$$SR_t(K) = \mathbb{E}[R(k)] \tag{15}$$

$$= \mathbb{E}[\mu(a_k^*) - \mu(a_k)] \tag{16}$$

$$= \mathbb{E}_{H_{k-1}}[\mathbb{E}[\mu(a_k^*) - \mu(a_k) \mid H_{k-1}]] \qquad \text{Note that } a_k \sim a_k^* \text{ when } H_{k-1} \text{ is fixed} \tag{17}$$

$$= \underbrace{\mathbb{E}[U(a_k, H_{k-1}) - \mu(a_k)]}_{\text{Part 1}} + \underbrace{\mathbb{E}[\mu(a_k^*) - U(a_k^*, H_{k-1})]}_{\text{Part 2}} \tag{18}$$

According to the properties shown in Eq. 7, the part 2 can be deducted:

$$\mathbb{E}[\mu(a_k^*) - U(a_k^*, H_{k-1})] \leq \mathbb{E}[(\mu(a_k^*) - U(a_k^*, H_{k-1}))^+]$$

$$\leq \mathbb{E}\left[\sum_{a_k}^{\mathcal{A}}(\mu(a_k) - U(a_k, H_{k-1}))^+\right]$$

$$= \mathbb{E}\left[\sum_{a_k}^{\mathcal{A}}(\mu(a_k) - U(a_k, H_{k-1}))^+\right] \tag{19}$$

$$= \sum_{a_k}^{\mathcal{A}} \mathbb{E}\left[(U(a_k, H_{k-1}) - \mu(a_k))^+\right]$$

$$\leq k \cdot \frac{\gamma}{\mathcal{K} \cdot k}$$

$$= \frac{\gamma}{\mathcal{K}}$$

Similarly, Eq. 7 suggests that the part 1 is bounded as follows:

$$\mathbb{E}[U(a_k, H_{k-1}) - \mu(a_k)] = \mathbb{E}[2r(a_k, H_{k-1}) + L(a_k, H_{k-1}) - \mu(a_k)]$$

$$= \mathbb{E}[2r_c(a_k, H_{k-1})] + \mathbb{E}[L(a_k, H_{k-1}) - \mu(a_k)]$$

$$\mathbb{E}[L(a_k, H_{k-1}) - \mu(a_k)] \leq \mathbb{E}[(L(a_k, H_{k-1}) - \mu(a_k))^+]$$

$$\leq \mathbb{E}\left[\sum_{a_k}^{\mathcal{A}}(L(a_k, H_{k-1}) - \mu(a_k))^+\right]$$

$$= \mathbb{E}\left[\sum_{a_k}^{\mathcal{A}}((\mu(a_k) - U(a_k, H_{k-1}))^+\right] \tag{20}$$

$$= \sum_{a_k}^{\mathcal{A}} \mathbb{E}\left[\mu(a_k) - L(a_k, H_{k-1}))^-\right]$$

$$\leq k \cdot \frac{\gamma}{\mathcal{K} \cdot k}$$

$$= \frac{\gamma}{\mathcal{K}}$$

Putting part 1 and Part 2 together, $SR_t = \frac{\gamma}{\mathcal{K}} + \frac{\gamma}{\mathcal{K}} + \mathbb{E}[2r(a_k, H_{k-1})]$. Summing up over t, it can be proved that $SR(\mathcal{K}) \leq 2\gamma + 2\sum_{k=1}^{\mathcal{K}} \mathbb{E}[r(a_k, H_{k-1})]$.

## B  Broader Impacts

In terms of the potential positive impacts, our work may have impacts on the improved efficiency and better decision-making processes based on LLMs with lower costs in many domain-specific tasks. The potential benefitted scenarios and group of people can be education, medical and healthcare for both research and application. We may also be able to reduce commercial costs for enterprises and innovations in the industry.

This paper does not present any foreseeable negative impacts to society since we focus on the technique of optimizing a model selection policy that is purely serving the KBQA task.

