# OpenReview forum: "Cost-efficient Knowledge-based Question Answering with Large Language Models"
_NeurIPS.cc/2024/Conference — NeurIPS 2024 poster_

### Official Review · Reviewer_dW52 · 2024-07-12

**Soundness:** 2
**Presentation:** 2
**Contribution:** 2
**Rating:** 5
**Confidence:** 3

**Summary:**

1. This paper proposes a cost-efficient strategy named "Coke" to automatically assign the most promising model for particular questions.
2. Experiments show the effectiveness of their method in Knowledge-based question answering (KBQA).

**Strengths:**

1. The problem definition and mathematical explanation is clear.
2. Experiments on 3 representative domain-specific datasets show the effectiveness of their method to improve performance and cost efficient.

**Weaknesses:**

1. As for the calls (times), I doubt it is a good metric because in generally users care more about call latency and commercial products always define token numbers to calculate price. (longer calls should have a larger price)
2. Maybe not only LLM cost but also KGMs' cost should be considered
3. 3 datasets are similar in the sense of reasoning, the generalizability is limited.

**Questions:**

See weakness

**Limitations:**

The authors discuss about the limitation of their work in one section.

---

> ### Author Rebuttal · Authors · 2024-08-07
>
> We gratefully thank you for your constructive comments which we believe will absolutely improve the quality of our paper. We also wish to invite you to check our new results with a more comprehensive consideration of evaluation metrics, cost of KGMs and generalizability.
>
> > Response to weaknesses
> * **W1: Evaluation metrics**.
> Thanks for your constructive comments, we are inspired and have included two more metrics **`Inference Latency`** and **`Cost Advantage`** to address your concerns. Indeed, in our submission, we have already included two metrics '**calls**' and '**API fees**' , where calls are used for evaluating open-source LLMs and API fees are for GPT series. We show the new results on three benchmark datasets as follows.
> **`Inference Latency (s)`**: the time used for making predictions in seconds.
> **`Cost Advantage (%)`**: as the percentage of questions answered by small models, while this metric is widely adopted for automatic ML and used in HybridLLM-ICLR'24.
>
> |                    |  Inference Latency (s)  | Cloud/API Fees ($) |  Inference Latency (s)  | Cloud/API Fees ($) |   Inference Latency (s)  | Cloud/API Fees ($) |
> |:------------------:|:-----------------------:|:------------------:|:-----------------------:|:------------------:|:------------------------:|:------------------:|
> |        HamQA       |          340.60         |        0.005       |          425.60         |        0.004       |          671.25          |        0.009       |
> |      GreaseLM      |          462.17         |        0.007       |          503.44         |        0.005       |          762.17          |        0.013       |
> |      Llama2 7B     |          61.20          |        0.20        |          60.00          |        0.20        |           61.20          |         0.4        |
> |      Llama3 8B     |          50.01          |        0.20        |          47.58          |        0.20        |           50.01          |         0.4        |
> |       GPT 3.5      |          26.33          |        0.05        |          27.29          |        0.02        |           26.33          |        0.15        |
> |        GPT-4       |          20.67          |        1.01        |          18.16          |        0.38        |           20.67          |        3.03        |
> |  Coke-HamQA (Ours) |          70.59          |       -20.16       |          58.25          |       -10.85%      |           46.12          |       -4.05%       |
> | Coke-Llama3 (Ours) |          36.22          |       -17.52%      |          25.37          |        -8.2%       |           30.41          |     **-41.92%**    |
> | Coke-Llama3 (Ours) | **CSQA Accuracy Imp%:** |       +2.48%       | **OBQA Accuracy Imp%:** |       +0.58%       | **MedQA Accuracy Imp%:** |       +3.26%       |
>
> | Cost Advantage (%) |  CSQA  |  OBQA  |  MedQA |
> |:------------------:|:------:|:------:|:------:|
> |  Coke-HamQA (Ours) | 20.89% | 11.02% |  4.32% |
> | Coke-Llama3 (Ours) | 18.62% |  9.70% | 48.55% |
>
> * **W2: Cost of KGMs**
> Thanks very much for the inspiration, following your suggestions, we have also considered and quantified the cost of local KGMs and local LLMs through **`cloud service fee`** in dollars. Comparisons have been made over three domain-specific datasets uniformly with the API fees of GPT series, where in this case, our evaluation is more convincing now. For details, please check the previous weakness for new results.
> **`cloud service fee`**: calculates the token-level cost based on basic requirements of GPU resources in one cloud server, instantiated by AWS g4dn.xlarge and p3.8xlarge with USD 0.526 and USD12.24 per hour.
> * **W3: Generalizability**
> Thanks for your constructive comments. We believe considering generalizability will definitely enhance the soundness of our paper and enlarge the impact to the community. To address your concerns, we would like to show our **`generalizability`** on two open-ended QA datasets, in addition to three original domain-specific multi-choice benchmarks. In this pipeline of setting, the models are not limited to providing answers with choices or under any pre-defined formats.
>
> |           (Hits@1)           | WebQSP |  CWQ  |
> |:----------------------------:|:------:|:-----:|
> |            KV-Mem            |  46.72 | 18.51 |
> |           EmbedKGQA          |  66.60 | 45.35 |
> |            GrafNet           |  66.35 | 36.72 |
> |            GPT 3.5           |  65.30 | 41.50 |
> |             GPT-4            |  80.58 | 60.42 |
> |          Coke-EmbedKGQA (Ours)         |  **86.47** | **61.83** |
> | Cost Sav. ($ Cloud/API Fees) | **30.21%** | **3.45%** |

---

> > ### Comment · Reviewer_dW52 · 2024-08-13
> >
> > Thanks for your response and detailed new results. After considering other reviewers' comments and your rebuttal, I increase my rating to accept level.

---

> > > ### Author Response · Authors · 2024-08-14
> > > **Grateful thanks to Reviewer dW52**
> > >
> > > Dear Reviewer dW52,
> > >
> > > We are so grateful for your recognition and agreement with other reviewers’ comments.
> > >
> > > We cherish your high-quality suggestions on metrics and KGM costs which were also raised by other reviewers. Your valuable suggestions have made our paper a much better one.
> > >
> > > We will keep revising our paper and include  all the results in the final version.
> > >
> > > Best regards,
> > > Submission 13626

---

### Official Review · Reviewer_JWNj · 2024-07-12

**Soundness:** 3
**Presentation:** 2
**Contribution:** 2
**Rating:** 6
**Confidence:** 3

**Summary:**

The paper introduces a method for deciding whether to use a Large Language Model (LLM) or a Knowledge Graph-based Model (KGM) to solve various Knowledge-based QA tasks in an episodic manner, based on historical data. The main goal is to achieve better performance at a lower cost throughout the entire QA process. This work formulates the problem as a multi-armed bandit problem and proposes a solution to this formulation.

**Strengths:**

- **Originality:** The problem addressed is interesting. The approach of using multi-armed bandit problem formulation to decide between the efficient use of small models leveraging KG and the effective use of LLMs is novel. There appears to be no prior work addressing this specific problem.
- **Quality:** The authors have appropriately formulated the problem and attempted to solve it using technically sound methods. The experimental results convincingly demonstrate the advantages of the proposed method.
- **Significance:** This work is likely to inspire future studies in system cost optimization especially for LLMs. The experimental results highlight the potential to reduce costs while improving overall performance, which is impressive.

**Weaknesses:**

- **Clarity:**
    - It is unclear if the cost in the experimental results is solely based on the number of times the LLM is used, with KGM usage considered cost-free. Additionally, the cost associated with using models like RoBERTa for Context-aware expert distinguishing, as discussed in Section 3.3, is not addressed in the experiments. Ignoring these local model costs might not be appropriate, and related discussions should be included in the experiments section.
    - The specifics of the dataset usage are also unclear. The results in Table 1 seem to imply that KGM's arm embedding was updated using fine-tuned train data and then tested directly on test data, but the experimental setup isn't clearly explained.
- **Significance:** There are questions regarding the practical applicability of the proposed method. The QA datasets used in the experiments are multiple-choice QA datasets, making accuracy measurement straightforward and benefiting from extensive research on KG-based models. However, applying this method in real-world applications like chat-bots poses challenges that need to be addressed. Despite this, the research lays a foundation for practical follow-up studies, which is a positive aspect.

**Questions:**

### Questions

1. Do you just set the cost of KGM to 0? I cannot find the related details in the experiments section.
2. In Figure 3, what is the difference between the three figures? Are they using different datasets?
3. In Figure 4, what is the value of the y-axis? Is it k, the number of iterations? What dataset is each figure representing?

### Suggestions

- L265: "While The" should be corrected to "While the".
- In Figure 4, the leftmost figure seems to have a typo in its y-axis label (000-1221 should be 1000-1221).

**Limitations:**

The authors already addressed the limitations in Section 6.

---

> ### Author Rebuttal · Authors · 2024-08-07
>
> We would like to sincerely express our gratitude for your encouraging support and strong recognition, especially for your emphasis of our contribution to the community in the weakness again. We will carefully revise the final version following your suggestions.
> > Response to weaknesses
> - **W1: Clarity**.
> Thanks for pointing this out, we will further clarify the setting in the revised version. In Section 2.2 Performance Evaluation, we introduced two metrics '**calls**' and '**API fees**', where calls (times) are used for evaluating our performance based on open-source LLMs and API fees (dollars\$) are for GPT series. In auto-ML research, small models are considered to be free since they are far cheaper than big models like LLMs.
> Following your suggestions, we have also considered and quantified the cost of local KGMs and local LLMs through **`cloud service fee`** in dollars. Comparisons have been made over three domain-specific datasets uniformly with the API fees of GPT series, where in this case, our evaluation is more convincing now.
> **`cloud service fee`**: calculates the token-level cost based on basic requirements of GPU resources in one cloud server, instantiated by AWS g4dn.xlarge and p3.8xlarge with 0.526 USD and 12.24 USD per hour.
>
> we have included two more metrics **`Inference Latency`** and **`Cost Advantage`** to comprehensively evaluate our performance. This also showcases that our evaluation can be general and followers can easily adapt to their own domain with specific evaluation metrics. We show the new results on three benchmark datasets as follows.
> **`Inference Latency (s)`**: the time used for making predictions in seconds.
> **`Cost Advantage (%)`**: used in HybridLLM-ICLR'24, as the percentage of questions answered by small models, while this metric is widely adopted for automatic ML.
>
> |                    |  Inference Latency (s)  | Cloud/API Fees ($) |  Inference Latency (s)  | Cloud/API Fees ($) |   Inference Latency (s)  | Cloud/API Fees ($) |
> |:------------------:|:-----------------------:|:------------------:|:-----------------------:|:------------------:|:------------------------:|:------------------:|
> |        HamQA       |          340.60         |        0.005       |          425.60         |        0.004       |          671.25          |        0.009       |
> |      GreaseLM      |          462.17         |        0.007       |          503.44         |        0.005       |          762.17          |        0.013       |
> |      Llama2 7B     |          61.20          |        0.20        |          60.00          |        0.20        |           61.20          |         0.4        |
> |      Llama3 8B     |          50.01          |        0.20        |          47.58          |        0.20        |           50.01          |         0.4        |
> |       GPT 3.5      |          26.33          |        0.05        |          27.29          |        0.02        |           26.33          |        0.15        |
> |        GPT-4       |          20.67          |        1.01        |          18.16          |        0.38        |           20.67          |        3.03        |
> |  Coke-HamQA (Ours) |          70.59          |       -20.16       |          58.25          |       -10.85%      |           46.12          |       -4.05%       |
> | Coke-Llama3 (Ours) |          36.22          |       -17.52%      |          25.37          |        -8.2%       |           30.41          |     **-41.92%**    |
> | Coke-Llama3 (Ours) | **CSQA Accuracy Imp%:** |       +2.48%       | **OBQA Accuracy Imp%:** |       +0.58%       | **MedQA Accuracy Imp%:** |       +3.26%       |
>
> | Cost Advantage (%) |  CSQA  |  OBQA  |  MedQA |
> |:------------------:|:------:|:------:|:------:|
> |  Coke-HamQA (Ours) | 20.89% | 11.02% |  4.32% |
> | Coke-Llama3 (Ours) | 18.62% |  9.70% | 48.55% |
>
> - **W2: Significance**.
> We appreciate your expertise and believe this could help us further increase the impact within the community. To address your concerns, we would like to show our **`generalizability`** on two open-ended QA datasets, in addition to three original domain-specific multi-choice benchmarks.
>
> |           (Hits@1)           | WebQSP |  CWQ  |
> |:----------------------------:|:------:|:-----:|
> |            KV-Mem            |  46.72 | 18.51 |
> |           EmbedKGQA          |  66.60 | 45.35 |
> |            GrafNet           |  66.35 | 36.72 |
> |            GPT 3.5           |  65.30 | 41.50 |
> |             GPT-4            |  80.58 | 60.42 |
> |          Coke-EmbedKGQA (Ours)         |  **86.47** | **61.83** |
> | Cost Sav. ($ Cloud/API Fees) | **30.21%** | **3.45%** |
> > Response to questions and suggestions
> - **Q1: Cost of KGMs**.
> Yes, as explained previously, small models are considered to be free in auto-ML since they are far cheaper than big models like LLMs. Following your suggestions, we have also considered and quantified the cost of local KGMs and local LLMs through **`cloud service fee`** in dollars\$.
> - **Q2: Figure 3 clarification**.
> Yes, three subfigures are drawn for CSQA, OBQA and MedQA respectively.
> - **Q3: Figure 4 clarification**.
> Yes, axis-Y represents for the intervals of selection times. This case study observes the selection as k goes, which showcases the ability of our framework to balance exploration and exploitation (color changes from shallow->deep or deep->shallow). The datasets are for CSQA, OBQA and MedQA respectively.
> - **S1&S2: Typos**.
> Thanks for your carefulness, we will fix the typos in the final version.

---

> > ### Comment · Reviewer_JWNj · 2024-08-13
> >
> > Thank the authors for their detailed response. The clarifications are clear, and most of my concerns have been addressed. Therefore, I maintain my original score.

---

> > > ### Author Response · Authors · 2024-08-14
> > >
> > > Dear Reviewer JWNj,
> > >
> > > We would like to sincerely and gratefully for your support again in the original comment for our contribution that we may inspire the follow-up work. Your encouragement has enlightened us to persist to make our work better during the rebuttal.
> > >
> > > We will keep revising the paper and include all results in the final version.
> > >
> > > Best regards,
> > > Submission 13626

---

### Official Review · Reviewer_FvaD · 2024-07-12

**Soundness:** 4
**Presentation:** 3
**Contribution:** 4
**Rating:** 8
**Confidence:** 4

**Summary:**

This manuscript presents a novel cost-efficient strategy to leverage LLMs for knowledge-based question answering. It could balance both inferential accuracy and cost saving. Several SOTA methods, inlcuding both traditional KGQA methods and LLMs are combined since KGQA models are small and knowledgeable but less accurate, while LLMs are comprehensive on general questions but rather expensive. Authors design a cluster-based TS technique and a tailored contextual MAB to filter out the experts and constrain the decision with the consideration on cost regrets.

**Strengths:**

This propsed method has fhe following strengths:

1. Saving costs of invoking LLMs while obtaining higher accuracy is a promising topic with both academic merits and industrial values. It could be somehow inspiring various research communities.

2. The proposed methodology is novel and reasonable. It considers the model selection based on three aspects: the accuracy potential of choosing one model based on historical success, the expertise on particular questions based on the question semantics and the cost regret from the expenditure on historical failure to control the costs.

3. Sufficient theoretical analysis and proofs, e.g., expectation bound of Thompson Sampling and the confidence bound for MAB to correspondingly support the design of automatic selection.

4. Nicely drawn running example is easy for grasping the motivation. It provides a sketched overview of the pipelines, the performance/cost comparison of existing methods and the overlaps among different models. This motivates the combination of KGMs and LLMs with a auto-selection algorithm.

5. Satisfying experimental performance on both accuracy and cost saving performance.

6. The writing is good and clear to follow.

**Weaknesses:**

1. A pseudo-algorithm should be provided to demonstrate the selection process and how the Thompson sampling and MAB decides in each step.

2. The results are currently from one fixed combination of base models, i.e., HamQA + ChatGPT+ GPT4. It will be more solid to provide the results of different combinations of base models as additional ablation studies. For example, I may like to see the results with GreaseLM + HamQA as the clustered arms for KGMs and ChatGPT+GPT as the ones for LLMs.

3. Missing references for ChatGLM Baichuan and Llama2/3 in the main table.

4. More heuristic baselines should be included, for instance, (1) $E_c$ + random $E_a$; (2) random $e_c$ + random $E_a$ (3) pure random selection without $E_c$, $E_a$ and $R_a$ (4) epsilon greedy-based selection with $E_a$ only.

**Questions:**

1: It will be interesting if the author can apply existing methods in other domain as baselines and make a comparison. Have the authors try the methods in other domains like FrugalGPT and HybridLLM and migrated them into KGQA?

2: The reviewer is interested to see more combinations of base models. Have you tried different combinations?

**Limitations:**

Yes.

The limitation is acknowledged by the authors, which is from the performance constraint by base models. It should be easy to address by replacing the base models with more advanced ones. While the selection of models requires much prior knowledge of performance, this may be a concern on additional resource consumption.

---

> ### Author Rebuttal · Authors · 2024-08-07
>
> We would like to gratefully thank you for your strong support! We also cherish your expertise for plenty of professional suggestions. It is encouraging to be acknowledged by experts from the community. We believe your insightful comments will greatly help us to enrich the experiments and further demonstrate our contributions.
>
> Following your suggestions, all the newly demonstrated algorithms and experiments will be added to the final revised version.
> > Response to weaknesses
> - **W1: Algorithms and pseudo-codes**.
> Thanks for your constructive suggestions. We have provided four pseudo-codes in the PDF of the general rebuttal, which demonstrates the training and inference stage of our main framework, as well as the cluster-based Thompson Sampling and contextual MAB. Please kindly check the details in the PDF.
> - **W2: Base model combinations**.
> We appreciate your insightful comments on a meaningful ablation study. We have done some preliminary trials during our experiments. Some intuitive results are shown hereunder over CSQA dataset. We observe that, both the quality and the number lay an indispensible influence on the final decision. As a fast-adaptive framework, we can readily include more advanced and cheaper base models to move the Pareto frontier for KGQA considering both higher accuracy and lower costs.
>
> |                                | 3       | 3 | 4       | 4 | 5       |
> |-----------------------------|-----|-----|-----|-----|-----|
> | HamQA                      | Picked |  -   | Picked |   Picked  | Picked |
> | GreaseLM                  | -         |  Picked   | Picked |   -  | Picked |
> | Llama3 (7b)               | -         |   -  | -         |  Picked   | Picked |
> | GPT 3.5                     | Picked |   Picked  | Picked |  Picked   | Picked |
> | GPT-4                        | Picked |  Picked   | Picked |  Picked   | Picked |
> | Coke (Ours)               | 2.74%  |   2.66%  |     1.89%     |   1.05%  |     0.41%     |
> | Cost Sav. (Cloud/API fees) | 20.16% |   21.50%  |    20.92%      |  19.84%   |     36.01%     |
>
>
> - **W3: Missing references**.
> Thanks for your carefulness, we have properly added citations for both LLMs.
> - **W4: Heuristic Baselines**.
> Thanks for your guidance, we have implemented the heuristic methods on all three domain-specific datasets. For epsilon-greedy, we have set the threshold as '0.7'.
>
> |          (Acc. Imp%)          |   CSQA  |   OBQA  | MedQA   |
> |:-----------------------------:|:-------:|:-------:|---------|
> |        E_c + Random E_a       | 2.21%   | 4.69%   | -0.4%   |
> | Cost Sav. (Cloud/API fees)    | 1.50%   | 0.38%   | 0.03%   |
> |   Random E_c and Random E_a   | -10.32% | -3.19%  | -15.88% |
> | Cost Sav. (Cloud/API fees)    | 1.50%   | 2.05%   | 12.41%  |
> |     pure random selection     | -62.51% | -45.27% | -84.40% |
> | Cost Sav. (Cloud/API fees)    | 64.48%  | 53.98%  | 70.25%  |
> | epsilon greedy-based MAB only | -12.79% | -22.54% | -12.79% |
> | Cost Sav. (Cloud/API fees)    | 9.76%   | 17.61%  | 8.06%   |
> |       Coke-HamQA (Ours)       |  2.74%  |  0.67%  | 1.03%   |
> | Cost Sav. (Cloud/API fees)    |  20.16% |  10.85% | 4.05%   |
> > Response to questions
> - **Q1: Migration Study from other domains**.
> Thank you for providing these two references. We will properly cite them in the related work section. For FrugalGPT, we set the threshold for switching KGMs to LLMs as '0.85' after necessary hyperparameter tuning.
>
> |           |  CSQA  |   OBQA  |  MedQA  |
> |:---------:|:------:|:-------:|:-------:|
> | HybridLLM | 1.26%  | -3.55%  | -19.98% |
> | FrugalGPT | -2.51% | -10.79% | -32.51% |
>
> - **Q2: Base model combinations**.
> Thanks for your valuable question. Yes, we have provided our observations and preliminary results in the aforementioned rebuttal.

---

> > ### Comment · Reviewer_FvaD · 2024-08-13
> > **Thanks for authors' comprehensive rebuttal.**
> >
> > Thanks for authors' comprehensive rebuttal. I like this paper after checking the new metric experiments suggested by other reviewers. I believe this paper for sure inspires much follow-up work.
> >
> > My concerns were all addressed. I hope the new results could be included in the final revision.
> >
> > In general, I still support the acceptance of this paper during the ac-reviewer discussion period and keep my positive score.

---

> > > ### Author Response · Authors · 2024-08-13
> > > **Grateful Appreciation to Reviewer FvaD**
> > >
> > > Dear Reviewer FvaD,
> > >
> > > We are deeply grateful for your strong support and recognition. Again, it is encouraging to be recognized by the experts from the community. We cherish the great suggestions from both you and other reviewers that significantly improve our paper.
> > >
> > > All the changes and new experiments will be discussed and included in the final version.
> > >
> > > Best regards,
> > >
> > > Submission 13626

---

### Official Review · Reviewer_HDrN · 2024-07-15

**Soundness:** 4
**Presentation:** 3
**Contribution:** 3
**Rating:** 7
**Confidence:** 3

**Summary:**

The authors proposed a strategy to switch between LLMs and KGMs when performing Question Answering, aiming to optimize both cost and accuracy. The evaluation is performed on three datasets: CommonsenseQA, OpenBookQA, and MedQA.

**Strengths:**

The authors address an important and practical problem, which is cost-saving in QnA. The introduction is well-written, and the intuition is nicely presented. The proposed idea to switch between KG-based models and LLMs depending on different questions makes sense and shows some promising results on the test datasets.

**Weaknesses:**

It lacks implementation details such as how the model is trained, which models are being used as LLMs and KGMs, and the exact formula/implementation of the cost functions.

The proposed solution is based on cluster-level Thompson Sampling, but it lacks an explanation of why it's the best candidate for this setting. Additionally, defining the cost function as the number of calls is less practical; instead, it should be a function of latency and API fees (assuming the use of GPT-4).

The result of cost-saving is promising but not as high as expected. From Figure 1c, it seems that about half of the questions can be answered by KGMs. However, the experimental cost-saving shown in Table 1 is only 10%-20%. This suggests there is still room for improvement in the decision-making model.

There is almost no benefit when using it on MedQA, so one weakness of the solution is that it heavily depends on the performance of the KGMs to save costs

**Questions:**

N/A

---

> ### Author Rebuttal · Authors · 2024-08-07
>
> Thanks for your recognition of our contributions to a very important and practical problem. We value your insightful comments and will carefully revise the final version following your suggestions.
>
> > Response to weaknesses
> * **W1: Implementation details**.
> Thanks for raising these points. In the original main result, we use HamQA, GPT 3.5 and GPT-4 as KGMs and LLMs.
> - *Framework training*. We provided two pseudo-codes in the PDF of general response to demonstrate the process. The optimization is under a Bayesian online learning framework where our model keeps learning with new queries before using up the budget and continuously updates the parameters based on historical observations. This enables us to be generalizable to any unknown scenarios.
> - *KGM pretraining*. We pretrain the base KGMs based on the training dataset. We carefully checked and ensured the data distributions among train, dev and test are identical. The pre-trained KGMs are directly leveraged for inference without further training.
> - *Cost functions*. We do not necessarily utilize a cost function since our model is optimized under Bayesian online learning. More generalizably, after defining the parameter prior, we calculated the posterior probability according to the feedback from LLMs. Instead of minimizing the cost function, we aim to optimize the expectation function and update the parameters including: 1. posterior distribution Beta($\alpha^{k−1}_{c}$, $\beta^{k−1}_{c}$) for clustered-Thompson Sampling 2. $\mu^{k−1}_{a}$ and $\eta^{k−1}_{a}$ for contextual MAB.
> * **W2-1: Mechanism of Thompson Sampling**.
> Thank you for the comment. We have added two pseudo-codes in the PDF of the general response to demonstrate: 1. clustered Thompson Sampling 2. contextual MAB.
>
> * **W2-2: Evaluation metrics**.
> Thanks for your constructive comments, we are inspired to have two more metrics **`Inference Latency`** and **`Cost Advantage`**. We also quantified the costs of KGMs and local LLMs with **`cloud service fee`** to address your concerns. In our submission, we have already included two metrics '**calls**' and '**API fees**', where calls are used for open-source LLMs and API fees are for GPT series. This also showcases that our evaluation can be general and easily adapted to various metrics. We show the new results on three benchmark datasets as follows.
> **`Inference Latency (s)`**: the time span between question input and prediction output in seconds.
> **`Cost Advantage (%)`**: used in HybridLLM-ICLR'24, as the percentage of questions answered by small models.
> **`cloud service fee`**: calculates the token-level cost based on basic requirements of GPU resources in cloud servers, instantiated by AWS g4dn.xlarge and p3.8xlarge with USD 0.526 and USD12.24 per hour.
>
> |                    |  Inference Latency (s)  | Cloud/API Fees ($) |  Inference Latency (s)  | Cloud/API Fees ($) |   Inference Latency (s)  | Cloud/API Fees ($) |
> |:------------------:|:-----------------------:|:------------------:|:-----------------------:|:------------------:|:------------------------:|:------------------:|
> |        HamQA       |          340.60         |        0.005       |          425.60         |        0.004       |          671.25          |        0.009       |
> |      GreaseLM      |          462.17         |        0.007       |          503.44         |        0.005       |          762.17          |        0.013       |
> |      Llama2 7B     |          61.20          |        0.20        |          60.00          |        0.20        |           61.20          |         0.4        |
> |      Llama3 8B     |          50.01          |        0.20        |          47.58          |        0.20        |           50.01          |         0.4        |
> |       GPT 3.5      |          26.33          |        0.05        |          27.29          |        0.02        |           26.33          |        0.15        |
> |        GPT-4       |          20.67          |        1.01        |          18.16          |        0.38        |           20.67          |        3.03        |
> |  Coke-HamQA (Ours) |          70.59          |       -20.16       |          58.25          |       -10.85%      |           46.12          |       -4.05%       |
> | Coke-Llama3 (Ours) |          36.22          |       -17.52%      |          25.37          |        -8.2%       |           30.41          |     **-41.92%**    |
> | Coke-Llama3 (Ours) | **CSQA Accuracy Imp%:** |       +2.48       | **OBQA Accuracy Imp%:** |       +0.58       | **MedQA Accuracy Imp:** |       +3.26       |
>
> | Cost Advantage (%) |  CSQA  |  OBQA  |  MedQA |
> |:------------------:|:------:|:------:|:------:|
> |  Coke-HamQA | 20.89 | 11.02 |  4.32 |
> | Coke-Llama3 | 18.62 |  9.70 | 48.55 |
>
> * **W3: Performance Improvements**.
> Thank you for initiating the discussion. First, cost savings between 10% and 20% is already satisfying industrial applications. For example, a typical intelligent customer service in large e-commercial company will handle around 3M tokens a day, after applying our framework, the cost saving can be remarkably around 6600 USD\$ a year. Second, in the table for W2, we showcase the performance of using cheaper LLMs (Llama3 7b) with GPT3.5 and GPT-4. When we adopted Llama3 to replace KGMs on MedQA, the performance was boosted with over 41.92% savings and 3.26% ACC improvements.
>
>
> * **W4: Importance of KGMs**.
> We are grateful and excited to claim that this limitation is no longer constraining our paper after being guided by all reviewers. First, this limitation can be easily solved since we are a fast-adaptive and pluggable framework. We can readily involve advanced KGMs given prosperous community to achieve higher performance. Second, this limitation can also be solved by using cheaper local LLMs than KGMs. Evaluated by cloud service fees, the results have been provided in the previous rebuttal, and we can achieve 41.92% cost savings while improving the accuracy around 3.26% on MedQA.

---

> > ### Comment · Reviewer_HDrN · 2024-08-13
> >
> > Thank you for addressing my comments.
> > A commonly thought-of approach to use is to train a classifier (e.g., BERT-based) on the input query. Could you discuss why Thompson Sampling is used instead?

---

> > > ### Author Response · Authors · 2024-08-13
> > >
> > > Dear Reviewer HDrN,
> > >
> > > We would like to gratefully thank you for your acknowledgment of addressing your concerns. We are also excited to discuss your comments on replacing TS with an embedding-based classifier. We will include the discussions in the final revision to highlight our contributions.
> > >
> > > Indeed, during the very first stage of feasibility investigation, we did consider training a classifier for model selection, e.g., MLP. We agree with you that this is an intuitive solution and worth discussing. We would like to highlight the contribution of our methods which combines **`TS`**, **`MAB`** and **`cost regret`**, and explain why an embedding-based classifier (e.g., BERT-based) hardly works from the following perspectives:
> > > - **Generalizability and Scalability**.
> > > Our framework is super flexible and could readily combine different base models or replace particular models with more advanced ones subject to different scenarios and domains. However, BERT-based classifiers fail to do so. First, they require more training data and will do an entire retraining when the number of base models increases or the particular base models are changed. Second, the training complexity significantly increases with the number of base models. This forces us to abandon embedding-based classifiers.
> > >
> > > Inspired by Reviewer FvaD, we show that we are able to readily change the combination of base models: 1. select different numbers of models 2. select different types of models.
> > >
> > > |                            | 3      | 3      | 4      | 4      | 5      |
> > > |----------------------------|--------|--------|--------|--------|--------|
> > > | HamQA                      | Picked | -      | Picked | Picked | Picked |
> > > | GreaseLM                   | -      | Picked | Picked | -      | Picked |
> > > | Llama3 (7b)                | -      | -      | -      | Picked | Picked |
> > > | GPT 3.5                    | Picked | Picked | Picked | Picked | Picked |
> > > | GPT-4                      | Picked | Picked | Picked | Picked | Picked |
> > > | Coke (Ours)                | 2.74%  | 2.66%  | 1.89%  | 1.05%  | 0.41%  |
> > > | Cost Sav. (Cloud/API fees) | 20.16% | 21.50% | 20.92% | 19.84% | 36.01% |
> > > - **Adaptability to unseen query types**.
> > > Our TS and MAB are inherently adaptive to unseen scenarios. If a new type of query is given, our TS and MAB can sufficiently update the posterior knowledge and explore which models perform best on this new type, allowing for a quick adjustment without full retraining.
> > > - **Exploration-Exploitation Trade-off**.
> > > We use TS and MAB to balance the exploration-exploitation process, to try out different base models and select the best-known model according to historical observations. However, BERT-based models will probably and consistently choose a model based on its pre-training knowledge without adapting to changes in performance over time.
> > > - **Consideration on cost saving**.
> > > Our model remarkably taking the cost-saving performance into consideration by evaluating the **`cost regret`** based on models' historical expenditure on failures, which cannot be realized by BERT-based models and also can hardly be combined since they are literally in different spaces, i.e., embedding space and probability space.
> > > - **Lack of labeled data**.
> > > As introduced in our first rebuttal, we are making decisions under an online learning framework which do not require any labeled data for training. It can adaptively learn from the feedback of the model selections (e.g., which model performed best for a given query) and update the posterior knowledge and distributions, making it more suitable in environments where labeled data is sparse or expensive to obtain. However, it becomes tremendously harder to train BERT-based embedding classifiers which require a significant amount of labeled data to perform well. The quality of the auto-selection would heavily depend on the quality and quantity of this data, which might not always be available.
> > > - **Efficiency of real-time decision making**.
> > > In real-world scenarios, efficiency or latency to make predictions matters a lot. Since TS and MAB are naturally designed for online learning, they could continuously update the posterior beliefs about which model is optimal as more data comes in. This makes it well-suited for systems that need to make rapid decisions.

---

> > > > ### Comment · Reviewer_HDrN · 2024-08-14
> > > >
> > > > Thank you for showing the flexibility of this framework when adding more models. The table above, along with another one mentioning scores on open-ended QA, has convinced me of the model's strengths. Please consider including that in the final revision. I have therefore changed my review to support acceptance of this work.

---

> > > > > ### Author Response · Authors · 2024-08-14
> > > > > **Deep gratitude to Reviewer HDrN**
> > > > >
> > > > > Dear Reviewer HDrN,
> > > > >
> > > > > We would like to express our deep gratitude to your recognition and we are super encouraging given your score raising.
> > > > >
> > > > > We are also excited to see the potential of our model after being inspired by you and all the other reviewers. All results will be further verified and included in the final version.
> > > > >
> > > > > Wish the prosperous community and we will continue to work hard for future work.
> > > > >
> > > > > Best regards,
> > > > > Submission 13626

---

### Author Rebuttal · Authors · 2024-08-07

General Response
We would like to sincerely thank all the reviewers for their valuable comments. We are also very excited to be highly acknowledged for our **`contributions`** and **`significance`** to future studies in a variety of communities. To address the concerns raised by reviewers, we have correspondingly added a range of new experiments, which we believe has made our approach much more comprehensive and convincing.

We wish to invite all reviewers to check our new results and observations, which will all be added in the final revised version:
1. We have additionally included 2 more metrics to evaluate our performance in terms of cost saving: **`Inference Latency`** (inspired by Reviewer HDrN and dW52) and **`Cost Advantage`**(inspired by Reviewer FvaD and HybridLLM-ICLR'24), while originally we already have '**calls**' for local LLMs and '**API fees**' for GPT series in the submission (now in total 4 metrics). This also showcases that our evaluation can be general and followers can easily adapt to their own domain with specific evaluation metrics.
2. We have also considered and quantified the cost of local KGMs and local LLMs through **` cloud service fee`** in dollars instantiated by AWS g4dn.xlarge and p3.8xlarge (inspired by Reviewer JWNj and dW52). Comparisons have been made over three domain-specific datasets uniformly with the API fees of GPT series, where in this case, our evaluation is more convincing now.
3. We further show our **`generalizability`** on two open-ended QA datasets (inspired by Reviewer JWNj and dW52) , in addition to three original domain-specific multi-choice benchmarks.
4. We have demonstrated more comparison results with different base model combinations and heuristic baselines.
5. Four pseodu-codes are provided to demonstrate the algorithm for our main framework, Thompson Sampling and MAB (inspired by Reviewer HDrN and FvaD).

Thank again for all reviewers' suggestions that have greatly improved our paper. For specific details, please kindly check the corresponding responses and the PDF.

---

### Decision · Program_Chairs · 2024-09-25

**Decision:**

Accept (poster)

**Comment:**

The paper proposes a novel strategy to optimise cost and accuracy in Question Answering (QA) tasks by dynamically switching between Large Language Models (LLMs) and Knowledge Graph-based Models (KGMs). This paper makes a solid contribution to the field of QA by proposing practical method for optimising cost and accuracy.

Generally, the reviews for this paper are positive: the authors have provided responses to the reviewers' concerns resulting in some score increases. Authors have clarified the implementation details and addressed the issue of evaluating costs by introducing new metrics and considering the cost of KGMs. There are several weaknesses, although the authors have provided reasonable responses:

* The initial submission lacked details regarding models used, the training process, and the exact formulation of cost functions. The authors have acknowledged this and plan to provide more details in the revised version.
* While the use of Thompson Sampling is justified, the paper could benefit from a more thorough explanation of why it is the most suitable choice for this setting.
*  Although the method shows promise, the actual cost savings achieved are somewhat lower than expected, particularly in the MedQA dataset. There is probably room for improvement in the decision-making model.
* The effectiveness heavily depends on the performance of the KGMs, which may limit its generalisability in scenarios where KGMs are less effective.

I would also suggest to the authors that tangential work studying cost-aware policy orchestration in LLMs could be another method to compare against in future papers: https://arxiv.org/pdf/2401.13979v1